# Autonomous Unmanned Aerial Vehicles in Search and Rescue Missions Using Real-Time Cooperative Model Predictive Control

**DOI:** 10.3390/s19194067

**Published:** 2019-09-20

**Authors:** Fabio Augusto de Alcantara Andrade, Anthony Reinier Hovenburg, Luciano Netto de Lima, Christopher Dahlin Rodin, Tor Arne Johansen, Rune Storvold, Carlos Alberto Moraes Correia, Diego Barreto Haddad

**Affiliations:** 1Drones and Autonomous Systems, NORCE Norwegian Research Centre, 9294 Tromsø, Norway; rune.storvold@norceresearch.no; 2Department of Engineering Cybernetics, Norwegian University of Science and Technology (NTNU), 7034 Trondheim, Norway; hovenburg@ieee.org (A.R.H.); cdahlin@ieee.org (C.D.R.); tor.arne.johansen@ntnu.no (T.A.J.); 3Graduate Program in Electrical Engineering (PPEEL), Federal Center of Technological Education of Rio de Janeiro (Cefet/RJ), Rio de Janeiro 20271-204, Brazil; luciano.netto@gmail.com (L.N.d.L.); profcarlosalbertocorreia@gmail.com (C.A.M.C.); diegohaddad@gmail.com (D.B.H.)

**Keywords:** UAV, path planning, unmanned aerial vehicles, search and rescue, model predictive control, particle swarm optimization, Ardupilot, DUNE, software in the loop, JSBSim

## Abstract

Unmanned Aerial Vehicles (UAVs) have recently been used in a wide variety of applications due to their versatility, reduced cost, rapid deployment, among other advantages. Search and Rescue (SAR) is one of the most prominent areas for the employment of UAVs in place of a manned mission, especially because of its limitations on the costs, human resources, and mental and perception of the human operators. In this work, a real-time path-planning solution using multiple cooperative UAVs for SAR missions is proposed. The technique of Particle Swarm Optimization is used to solve a Model Predictive Control (MPC) problem that aims to perform search in a given area of interest, following the directive of international standards of SAR. A coordinated turn kinematic model for level flight in the presence of wind is included in the MPC. The solution is fully implemented to be embedded in the UAV on-board computer with DUNE, an on-board navigation software. The performance is evaluated using Ardupilot’s Software-In-The-Loop with JSBSim flight dynamics model simulations. Results show that, when employing three UAVs, the group reaches 50% Probability of Success 2.35 times faster than when a single UAV is employed.

## 1. Introduction

Search and Rescue (SAR) is one of the fields where the employment of UAVs brings many advantages over manned missions, such as its reduced costs, lower use of human resources, and mental and perception limitations of human operators. Reference [1] was one of the first works to perform experimental tests of a complete autonomous single UAV SAR solution. A probability density function (PDF) that expressed the likelihood of the target’s location was one of the main inputs of the system. Video data from the UAV was transmitted to the ground station that processed it in real time using computer vision techniques to detect the presence of the target and update the PDF. Paths were generated by the ground station to maximize the probability of finding the targeted object. The experimental flights showed satisfactory results in searching and detecting the target. The main necessary improvements identified by the authors were to implement on-board computing and to use multiple UAVs in the future.

SAR missions with autonomous UAVs are usually defined as an exploration problem. Exploration approaches can be used in a wide range of applications. For example, ice management, such as proposed by [2], where a Centralized Model Predictive Search Software was used for surveillance and tracking of ice using multiple UAVs. In the solution proposed by this reference, the algorithm finds a set of optimal waypoints that are sent to the autopilot control unit. The solution was tested in a Software-In-The-Loop environment and the results were evaluated for a different number of UAVs. In reference [3], the authors compared five exploration algorithms for SAR missions using a team of multiple UAVs. A centralized mission planner was used for the path-planning, whose objective was to find a sequence of waypoints for each UAV to follow. In [4], an offline algorithm to find a sequence of waypoints to be followed by a team of UAVs in a SAR mission is proposed. The cost function is built to minimize the travelled distance and the risk that people are exposed to. The authors created a non-uniform “risk map” and “probability of people map” for the simulated scenario.

A broad literature review about the persistent surveillance problem was done by [5] focusing on the use of multiple UAVs. Persistent surveillance is a type of exploration problem where the areas must be revisited over time. Among the many topics that the literature review covers, grid decomposition and path-planning techniques are the ones of the most interest for this work. The author reviews the most common types of grid decomposition classifying the rectangular one that is also used by the enhanced solution proposed by this paper, as the most popular. Regarding the path-planning techniques, the author states that the most common methods are classical search methods such as A* [6], decision theoretic methods such as Mixed Integer Linear Programming (MILP) [7], and Spanning Tree Coverage (STC) methods [8]. Model Predictive Control (MPC) [9] is mentioned as a topic less studied compared to the other planning techniques, but with significant advantages because it directly incorporates dynamic constraints, it is less heuristic and can react to changes in the environment.

MPC is a receding horizon control technique where the motion constraints are integrated in the control problem, which is particularly interesting for problems with fixed-wing UAVs. In addition, as the optimization is done for a finite time horizon, the technique is proper for real-time problems where the environment can dynamically change during the mission execution. In [10], MPC was used for sea Search and Track missions using an autonomous UAV. Hardware-In-The-Loop tests were performed. Waypoints were optimized and sent to the autopilot. Gimbal attitude was also optimized and sent to the servo system. The MPC optimization was not run on-board but on a dedicated computer in the ground control station. In [11], an MPC solution was proposed for target tracking using a solar-powered UAV. Energy harvesting and consumption were also included in the cost function. A three-dimensional dynamic model was used with the thrust, roll angle and lift coefficient as control inputs. The model also included horizontal winds. In reference [12], the authors describe a tracking application using multiple solar-powered UAVs in urban areas. A two-dimensional kinematic model is used without considering the wind. The cost function includes the energy harvesting and considers the shadows made by the buildings. Energy consumption is also considered. In addition, a distance constraint was included to guarantee the communication quality. The cooperative solution used is similar to the one described in this study, where the UAVs share their planned states. The meta-heuristic technique Grasshopper Optimization Algorithm is used for the optimization. A performance comparison between optimization techniques is presented, including the performance of Particle Swarm Optimization algorithm, which is used in this study. In this reference, the cost function is composed by the feasibility cost, the mission cost, the energy cost, the safety cost, the collision cost and the communication cost. In reference [13], the authors proposed a decentralized MPC for formation flight with obstacle avoidance. The chosen formation strategy was to keep a desired distance and angle between the UAVs and a moving reference point representing a UAV that follows the reference trajectory. Collision avoidance was also included in the cost function combined with a priority strategy where lower priority UAVs treat other UAVs as obstacles if they are within the protected radius. A two-dimensional UAV kinematic coordinated turn model without considering the wind was used. Numerical simulations were done using MATLAB^®^, where flight dynamics simulations neither control system software closed loop were considered. In [14], a collision avoidance decentralized MPC solution was proposed for multiple wheeled vehicles. External disturbances were also included in the form of translational, rotational, and speed alterations. A high level trajectory planning module was responsible for finding the planned path based on a nominal model and a low level trajectory tracking module was responsible for ensuring that the vehicles follow the planned trajectory facing the disturbances. MATLAB^®^ simulations were done for a wide variety of scenarios and number of vehicles and experiments were done using two robots in order to show real-time implementation. In [15], a similar strategy of dual control MPC was used. The solution was applied to a scenario with a single robot that aims to reach a given destination. Static and dynamic obstacles were simulated and measurement noise and process disturbance were included. Performance and real-time implementation were tested by simulations and experiments with a ground mobile robot. In [16], a cooperative multiple UAVs solution using MPC was used to close the communication link between a moving Autonomous Surface Vehicle and the ground station. Each UAV had to minimize a local cost function that took into consideration the planned states of the adjacent UAVs. In [17], a multiple UAVs receding horizon strategy was proposed for a cooperative surveillance problem. A potential field method was used for collision avoidance and network topology control management. The cooperative searching model was established based on the detection probability of the UAVs on targets in cells. In addition, a forgetting factor was included to indicate how fast the detection efforts are forgotten, so the UAVs can revisit the areas that were searched before. Simulations for different parameters were compared. In addition, the performance of the proposed method was compared to the performance of a parallel sequence search. In [18], a multi-vehicle cooperative search solution was proposed using MPC. Decoupled, centralized, cooperative and greedy approaches were compared.

In this study, a multiple UAVs cooperative Nonlinear Model Predictive Control solution to search a given area is proposed. The technique of Particle Swarm Optimization is used to find the control inputs that solve the optimal control problem. A coordinated turn kinematic model is implemented considering the effects of wind. The search area is divided into cells and each cell has an associated reward that in this work is defined according to the international Search and Rescue directives. A Software-In-The-Loop (SITL) environment with flight dynamics simulations is used to test the solution. SITL simulations are more realistic numerical simulations that run in real-time and closed loop with the control system software, in which aircraft sensor data are simulated by a flight dynamics model.

In addition, it is assumed that there is always connection between UAVs in the search area so that they can always share information between them. It is beyond the scope of this work to discuss network design. Research about communication and networks for UAVs in maritime missions is available at [19,20], where communication technologies, standards and protocols are discussed and system architectures are proposed, including hardware description.

The main contributions of this research to the field are:(1)Research about the use of receding horizon techniques in exploration problems is limited. This research contributes to the field by bringing a novel multiple UAVs solution to an exploration problem using Model Predictive Control where a finite time horizon grid search cost function with cells rewards and terminal cost is proposed.(2)The solutions found in the literature are developed with simplified vehicle models, in which the effects of wind are not considered when UAV platforms are employed. In small UAV missions, wind can easily exceed half of the UAV’s airspeed, significantly affecting the UAV performance. In this work, a coordinated turn kinematic model that takes the wind into consideration is designed. The model is developed so that the control inputs are the same as the autopilot control unit controls.(3)Usually, the solutions in the literature are only simulated in environments without embedded programming restrictions and where vehicle dynamics are not simulated. This makes results not close enough to what is expected in real-life applications. Implementing the solution in software for real-time applications brings additional challenges such as communication delays, processing time, actuator limitations, among others. In this research, the algorithm is fully implemented in an embedded software and tested in a real-time SITL environment that also simulates the flight dynamics, bringing results that are very close to reality.(4)In the literature, the SAR scenarios created for testing the proposed solutions are not based on standard directives. Therefore, making a fair comparison between different solutions is difficult. In this work, international SAR directives are considered to define the mission scenario and the performance indicators. This also allows for testing the proposed solution in a relevant case.

## 2. Materials and Methods

### 2.1. Embedded System

The path-planning algorithm was implemented as a task in DUNE: Unified Navigation Environment [21]. It allows the operation of a wide variety of robots using the same environment. This facilitates the development because the communication between DUNE and the different control units is transparent to the user.

The embedded system is outlined in Figure 1. The communication between DUNE and the Ardupilot [22] autopilot control unit is done via MAVLink Micro Air Vehicle Protocol [23]. To command the Ardupilot, DUNE’s MPC task must dispatch a message with the desired command, which will be interpreted by DUNE’s Ardupilot control task and then sent to the Ardupilot via MAVLink.

The communication between DUNE tasks is done via the IMC: Intermodule Communication API protocol [21]. This protocol basically operates by dispatching and consuming messages. Thus, if a message is dispatched by a task, another task that is waiting for that message will consume it.

The communication with other UAVs in DUNE’s cloud and with the ground station is also done via IMC protocol. The vehicles must be in the same network. Therefore, IP addresses and TCP and UDP ports must be properly configured and included in each vehicle’s DUNE configuration files. To allow the communication between vehicles, the InterVehicleCommon UDP transport module is used.

The MPC task is inside DUNE. The commands to control the UAV are given by the DesiredSpeed and DesiredRoll IMC messages, which carry, respectively, the airspeed and roll control inputs given by the optimization. These messages are interpreted by DUNE’s Ardupilot control task that sends the correspondent MAVLink message to the Ardupilot unit. In addition, DUNE’s Ardupilot control task is responsible for changing the Ardupilot flight mode to the Fly-By-Wire-B (FBWB) mode when the mission is started. In this flight mode, the Ardupilot control unit is responsible for holding the aircraft’s altitude and compensating the loss of lift caused by the rolling. Airspeed, roll angle and altitude external controls are accepted by the Ardupilot in the FBWB mode. The Ardupilot control unit is responsible for the low level control loops to maintain the commanded altitude, airspeed and roll angle. In this work, only roll and airspeed are controlled by the optimization algorithm; therefore, the altitude control is fixed at the desired altitude during the whole mission.

DUNE’s Ardupilot control task is also responsible for receiving the UAV’s pose and attitude information and dispatching it in the IMC messages EstimatedState and IndicatedSpeed. These messages are consumed by the MPC task to be used as the current state of the UAV.

The communication between the UAVs is done by the multiagent message, which was created and included in the IMC messages list specifically for this application. This message carries the information that the UAVs need to share between them, such as planned control inputs and current state.

Each UAV waits for the multiagent messages from all other UAVs before running the MPC optimization, e.g., if a team of three UAVs is employed, UAV 0 only runs the optimization algorithm after receiving the multiagent messages with the needed information from UAVs 1 and 2. This flow is described in Figure 2. Once the messages are received and the optimization loop is over, the UAV dispatches its multiagent message containing all information that need to be shared with the other UAVs.

In order to prevent control issues caused by communication and processing delays, the UAV states used by the MPC optimization are predicted according to the previously commanded control inputs and the measured delays by using Equation (Equation 7). Therefore, in the end of the optimization loop, the real states of the UAVs are expected to be close to the ones considered by the optimization algorithm when finding the optimal control inputs. This results in a superior behavior since the control inputs are obtained for UAVs attitudes and positions that are closer to reality.

Note that, in an extension of the method proposed by this work, measures can be employed to protect the system from communication failures, so that the UAVs do not wait for delayed messages for too long but use predictions instead. In addition, distance constraints can be included in the MPC in order to guarantee that the UAVs do not separate further than the maximum distance that enables the communication.

### 2.2. Optimal Control Problem

#### 2.2.1. Coordinated Turn Model

A two-dimensional kinematic model was developed based on the Coordinated Turn model for level flight [24]. In this model, the UAV turns by changing its roll angle so that there is no net side force acting on the UAV. As the effects of the increased angle of attack to compensate the lift losses when rolling do not have a significant impact on the kinematics, they are not included in the model for simplicity. Therefore, it is possible to relate the course rate and the roll angle by making the centrifugal force acting on the UAV equal and opposite to the horizontal component of the lift force acting in the radial direction (Figure 3):(1)$$F_{\mathrm{lift}}\sin u_{\phi }\cos\left(\chi -\psi \right)=mv_{g}\dot{\chi },$$
where $F_{\mathrm{lift}}$ is the lift force in [N], $u_{\phi }$ is the roll control input in [rad], χ is the course angle in [rad], ψ is the aircraft heading in [rad], *m* is the aircraft mass in [kg], $v_{g}$ is the ground speed in [m/s] and $\dot{\chi }$ is the course rate in [rad/s].

In addition, the vertical component of the lift force should be equal and opposite to the projection of the gravitational force (Figure 3):(2)$$F_{\mathrm{lift}}\cos u_{\phi }=mg,$$
where *g* is the gravitational acceleration of 9.81 m/s^2^.

By combining Equations (Equation 1) and (Equation 2), it is possible to find the relation between the course rate and the roll control input:(3)$$\dot{\chi }=\frac{g}{v_{g}}\tan u_{\phi }cos\left(\chi -\psi \right).$$

As wind is a major issue on UAV missions as it can likely reach more than half of the UAV’s maximum airspeed, the Coordinated Turn model used in this project was developed to consider the influence of wind on the UAV kinematics.

Therefore, the coordinated turn kinematic model for level flight in the presence of wind is given by:(4)$$\left(\begin{array}{c} \dot{x}\\ \dot{y}\\ \dot{\chi } \end{array}\right)=f\left(x,u\right)=\left(\begin{array}{c} v_{g}\cos\chi \\ v_{g}\sin\chi \\ \frac{g}{v_{g}}\tan u_{\phi }cos\left(\chi -\psi \right) \end{array}\right),$$
where x=(x,y,χ) are the north and east positions in the NED (North-East-Down) frame in [m] and the course angle in [rad], respectively. $u=(u_{v},u_{\phi })$ are the airspeed control input in [m/s] and roll control input in [rad], respectively, and with the ground speed ($v_{g}$ in [m/s]):(5)$$v_{g}=\sqrt{{\left(u_{v}\cos\psi +v_{w}\cos\psi_{w}\right)}^{2}+{\left(u_{v}\sin\psi +v_{w}\sin\psi_{w}\right)}^{2}},$$
where $v_{w}$ is the wind speed in [m/s], $\psi_{w}$ is the wind heading in [rad] and with the aircraft heading (ψ in [rad]) calculated using the law of sines:(6)$$\psi =\chi -\arcsin\left(\frac{v_{w}}{u_{v}}\sin\left(\psi_{w}-\chi \right)\right).$$

The model is discretized by the forward Euler method:(7)$$x_{k+1}=f_{d}\left(x_{k},u_{k}\right)=x_{k}+T_{s}f\left(x_{k},u_{k}\right),$$
where $x_{k}$ and $u_{k}$ are the state and control inputs vector, respectively, in the discretization step *k* and $T_{s}$ is the sampling period.

#### 2.2.2. Model Predictive Control Problem

To reach the mission goal, a centralized optimization approach might not be feasible because the problem would be too complex with too many control inputs. In a non-convex problem with a very long vector of variables to optimize, falling very early in a local minimum is a common issue. In addition, the necessary processing power to optimize so many control inputs would be difficult to achieve by the on-board processing unit of the UAV. In contrast, optimizing the controls of all UAVs in a ground station would not be an ideal solution, due to communication range limitations and because, in case of a communication failure, the UAVs would not receive its controls, which could compromise the mission.

Therefore, in this research, the problem is addressed as a cooperative control problem, where each UAV optimizes its own control inputs to update its state so that a local cost function is minimized. The cost function also takes into consideration the planned states of the other UAVs. As each UAV follows the same process, it is expected that the global mission goal is achieved cooperatively. Collision avoidance between UAVs is also considered.

Considering *I* UAVs ($x^{i},\forall i\in \left\{0,\ldots ,I-1\right\}$), the algorithm finds a control input sequence $U_{k}^{i}=\left\{u_{0}^{i},u_{1}^{i},\ldots ,u_{K-1}^{i}\right\}$ ∈ $R^{2\times K}$ for the *i*th UAV, which solves the following optimal control problem: (8)$$\begin{array}{cc} \mathrm{minimize}\;\; & \delta \left(C{\bar{x}}_{K}\right)+\sum_{k=0}^{K-1}L^{i}\left(C{\bar{x}}_{k},u_{k}^{i}\right), \end{array}$$(9)$$\begin{array}{cc} \mathrm{subject}\;\mathrm{to}\;\; & x_{k+1}^{i}=f_{d}\left(x_{k}^{i},u_{k}^{i}\right), \end{array}$$(10)$$\begin{array}{cc} \; & v_{a_{min}}\leq u_{v_{k}}^{i}\leq v_{a_{max}}, \end{array}$$(11)$$\begin{array}{cc} \; & \phi_{min}\leq u_{\phi_{k}}^{i}\leq \phi_{max}, \end{array}$$(12)$$\begin{array}{cc} \; & |C(x_{k}^{i}-x_{k}^{j})|>r_{c},\forall j\in \left\{0,\ldots ,I-1\right\}\backslash \left\{i\right\}, \end{array}$$
where
(13)$$\delta \left(C{\bar{x}}_{K}\right)=F\left(Cx_{K}\right)-aJ\left(C{\bar{x}}_{K}\right),$$
and
(14)$$L^{i}\left(C{\bar{x}}_{k},u_{k}^{i}\right)=aJ\left(C{\bar{x}}_{k}\right)+b{\left(u_{v_{k}}^{i}-u_{v_{k-1}}^{i}\right)}^{2}+c{\left(u_{\phi_{k}}^{i}-u_{\phi_{k-1}}^{i}\right)}^{2}.$$

Consider $u_{v_{-1}}^{i}$ and $u_{\phi_{-1}}^{i}$ as the commanded airspeed and roll angle for the *i*th UAV, respectively, in the previous optimization loop, ${\bar{x}}_{k}=\left[x_{k}^{0},\ldots ,x_{k}^{I-1}\right]$ as the states of all UAVs, *K* as the number of horizon steps and $r_{c}$ as the minimum safe distance between the UAVs to avoid collision. *a*, *b*, *c* are constant weighting factors and $C\in R^{2\times 3}$ is used to define that only the *x* (north) and *y* (east) positions are used from the state vector: (15)$$C_{1}=\left[\begin{array}{ccc} 1 & 0 & 0\\ 0 & 1 & 0 \end{array}\right].$$

In Equation (Equation 13), the function *J* represents the grid search function, which is the sum of the rewards of visited cells, and *F* is the terminal cost (cost-to-go) function, which is the distance from the terminal position to the unvisited cell with highest reward. Both functions are described in detail in Section 2.4.

#### 2.2.3. Optimization Technique

In this application, the Particle Swarm Optimization (PSO) [25] technique is used to find an optimal set of airspeed ($u_{v}$) and roll angle ($u_{\phi })$ that minimizes the cost function. PSO is a meta-heuristic optimization method where the particles (solutions) are updated every iteration based on the best global and local solutions. In this application, a standard PSO algorithm was implemented using CUDA C programming language in order to benefit from the parallelism of the NVIDIA^®^ Graphics Processing Unit that is assumed to be used in the UAV on-board computer.

The algorithm was set to run a fixed number of iterations on every loop. In addition, the number of particles must be defined. These two parameters affect the processing time and need to be fine-tuned according to the requirements.

The initial solutions are initiated with random values following the uniform distribution, where the minimum and maximum values are the defined boundaries of the airspeed and roll angle control inputs (Equations (10) and (11)).

### 2.3. SAR Directives Applied to UAVs Equipped with Remote Sensing

The Search and Rescue (SAR) consists, according to the Department of Defense (DoD) of the United States of America, in “the use of aircraft, surface craft, submarines, and specialized rescue teams and equipment to search for and rescue distressed persons on land or at sea in a permissive environment” [26]. This work focuses on the sea cases; therefore, the following description emphasizes sea SAR missions. In addition, as only Unmanned Aerial Vehicles (UAVs) are used in this work, only the directives for aircraft facilities are studied.

#### 2.3.1. Search Area

According to the International Aeronautical and Maritime Search and Rescue (IAMSAR) Manual [27], the Total Adjusted Search Area ($A_{t}$), which is the mission’s actual search area, is calculated based on the Total Available Search Effort ($Z_{ta}$), the Optimal Search Area ($A_{o}$) and the targeted Probability of Detection (POD). The first is a measure of the total area that a set of search facilities can effectively search within limits of search speed, endurance, and sweep width. The second is the search area which will produce the highest probability of success when searched uniformly with the available search effort and is essentially calculated based on the leeway and the Datum probable position error. Leeway is the the movement of a search object through water caused by winds blowing against exposed surfaces and Datum is a geographic point, line, or area used as a reference in search planning, such as the “Last Known Position” or the “Estimated Incident Position”.

If the Total Available Search Effort ($Z_{ta}$) is smaller than the Optimal Search Area ($A_{o}$), a strategy must be chosen to balance the Probability of Detection (POD) and the Total Adjusted Search Area ($A_{t}$). Usually, the chosen strategy is to fly on higher altitudes, increasing the sensor’s footprint or the crew’s field of view while decreasing the POD. However, in this work, as UAVs equipped with automated remote sensing are assumed to be used, resolution requirements usually can not be relaxed. Therefore, no trade-off between the POD and the search area is made and the POD is set to its maximum value of one, which makes the Total Adjusted Search Area ($A_{t}$) equal to the Total Available Search Effort ($Z_{ta}$).

In order to calculate the Total Available Search Effort ($Z_{ta}$), the sweep width (*W*) must be defined. When employing UAVs equipped with automated remote sensing in such missions, the sensor being used has a direct influence on this parameter. Altitude, view angle and image quality may affect the capability of identifying a survivor or an object on the sea. This is especially important to be taken into account because, if the image does not contain the object of interest properly recorded, the computer vision algorithm will not identify it, independently of the ability that the algorithm has on identifying an important occurrence on an image. This can occur due to low image quality or too long sensing distances, making the object of interest imperceptible.

In the IAMSAR manual, the sweep width is calculated based on the altitude of the aircraft, the visibility and the sensor system specifications. In sea SAR missions with aircraft facilities and visual search, the Corrected Sweep Width (*W*) is adjusted regarding the weather, velocity and crew fatigue correction factors. However, these factors can be excluded from the equation when automated remote sensing systems are used and the system’s velocity constraints are respected. Therefore, in this work, the Corrected Sweep Width (*W*) is considered equal to the original Uncorrected Sweep Width ($W_{u}$).

Finally, the Search Effort ($Z_{a}$), which represents the area which can be covered by a specific facility, is calculated by:(16)$$Z_{a}=V\times T\times W,$$
where *V* is the Search Facility Speed (average speed) in [m/s], *T* is the Search Endurance in [s] and *W* is the Sweep Width in [m].

Note that the Search Endurance is the time available for the facility to fly looking for the survivors. The IAMSAR manual considers this time as 85% of the lower value between the Daylight Hours Remaining and the On-Scene Endurance. This is due to the fact that human crew is often only able to search with visible light. In contrast, UAVs are often capable to equip sensors that are not affected by that, such as infrared cameras, which allows the task to be done even along the night. This is a considerable advantage of using UAVs equipped with remote sensing systems.

By summing the Search Effort of all facilities, the Total Available Effort ($Z_{ta}$ in [m^2^]) can be found:(17)$$Z_{ta}=\sum_{f=1}^{F}Z_{af},$$
where *F* is the number of facilities.

As described above, in this work, the Total Adjusted Search Area ($A_{t}$) is equal to the Total Available Effort ($Z_{ta}$). Therefore, for Single Point Datum, the Length and the Width of the search area are given by the square root of the Total Available Effort ($Z_{ta}$) as defined by the IAMSAR manual.

#### 2.3.2. Probability Map

The Probability of containment (POC) distribution in the search area is very important to guarantee efficient employment of the SAR facilities. When the initial indications do not provide enough information about the area, a standard distribution is assumed. The two most used types of standard distributions are the standard normal distribution and the uniform distribution, according the nature of the datum. For datum point and lines, the standard normal distribution is used. For datum areas, the uniform distribution is the most used. In this work, only the single point datum case is studied. Single point datum occurs, for example, when there is no significant leeway (e.g., when the target is a person in water [28]).

The probability map is a set of grid cells where each cell is labelled with the probability of containing (POC) the search object in that cell. As the the probability map follows a probability distribution function, the total sum of all cells should be equal to 100%. An example of probability table for single point datum with 5×5 cells is shown in Figure 4.

### 2.4. Cost Function

An exploration cost function was developed based on [18] to search a given area.

The region of interest is divided into M×N square cells of a width ($r_{e}$ in [m]), whose value must be chosen to be smaller than the optical imaging sensor’s footprint radius ($R_{e}$ in [m]) times the square root of 2. The sensor radius is equal to the radius of the circle inscribed in the sensor’s footprint. Figure 5 shows an example of a 4×4 grid with 100 m of cell width ($r_{e}$) and a UAV at position Cx equipped with a sensor with 100 m of radius ($R_{e}$).

The matrix $B^{i}$ ∈ $R^{M\times N}$ is used to identify if a cell was visited by the *i*th UAV. The matrix $b^{i}$ ∈ $R^{M\times N}$ is used to identify if a cell is planned to be visited by the *i*th UAV in the MPC horizon. In every M×N matrix used to identify if the cells are visited, each element has an associated value of 1 if the referring cell is visited or 0 if it is unvisited. Each cell has also an associated reward, given by ϕ ∈ $R^{M\times N}$.

The function $J({\bar{x}}_{k})$ is the sum of all cells associated value (1 or 0) in the step *k* times the correspondent reward:(18)$$J\left({\bar{x}}_{k}\right)=\sum_{m=0}^{M-1}\sum_{n=0}^{N-1}\phi_{mn}y_{{mn}_{k}}\left({\bar{x}}_{k}\right),$$
with
(19)$$y_{{mn}_{k}}\left({\bar{x}}_{k}\right)=(∥Cx_{k}-r_{mn}^{1}∥<R_{e}\wedge ∥Cx_{k}-r_{mn}^{2}∥<R_{e}\wedge ∥Cx_{k}-r_{mn}^{3}∥<R_{e}\wedge ∥Cx_{k}-r_{mn}^{4}∥<R_{e})\vee y_{{mn}_{k-1}},$$
where $r_{mn}^{1}$, $r_{mn}^{2}$, $r_{mn}^{3}$ and $r_{mn}^{4}$ are the four vertices of the cell (Figure 5) and $y_{{mn}_{k-1}}$ is the associated value of the cell in the previous horizon step.

The starting value of $y_{{mn}_{0}}$ is given by the logical sum of the matrices of already visited cells of all UAVs and the matrices of cells planned to be visited by other UAVs:(20)$$y_{{mn}_{0}}=B_{mn}^{i}\vee B_{mn}^{j}\vee b_{mn}^{j},\forall j\in \left\{0,...,I-1\right\}\setminus \left\{i\right\}.$$

Finally, $F(x_{K})$ is the terminal cost. This function is necessary for the algorithm to consider the search beyond the prediction horizon by having a *cost-to-go* term. It is given as the minimum Euclidean distance from the latest state of the UAV in the horizon, to the center of the closest unvisited cell, weighted by the correspondent reward, in the end of the horizon:(21)$$F\left(Cx_{K}\right)=\min_{\forall m\in O,\forall n\in P}\frac{∥Cx_{K}-r_{{mn}_{K}}∥}{\phi_{mn}},$$
where O⊆M and P⊆N are subsets of all unvisited cells and r=[x,y] are the north and east positions of the cell’s center.

## 3. Results and Discussion

### 3.1. Software-In-The-Loop Simulations Environment

To evaluate the proposed solution, a Software-In-The-Loop (SITL) environment was set up using Ardupilot SITL simulator. This simulator allows for testing the behavior of the Ardupilot code by running the Ardupilot in any computer without the Ardupilot hardware. The aircraft sensor data are simulated by JSBSim [29], an open source Flight Dynamics Model. Therefore, it is able to compute the UAV dynamics according to the actuator controls given by the Ardupilot code.

An aircraft platform model must be chosen for JSBSim flight dynamics calculations. In this work, the Skywalker X8 UAV platform (Figure 6) is used. The X8 is a UAV with around 4.0 to 4.5 kg of weight including the payload, 2.1 m of wingspan and 35.7 cm of mean aerodynamic chord. The aerodynamic model used to feed the JSBSim configuration file was developed based on wind tunnel tests [30]. In addition, the JSBSim source code was modified to use in its calculations the same wind map that is used by the MPC optimization.

Figure 7 shows the interconnection between modules. For each UAV, an Ardupilot SITL instance must be started linked to a JSBSim module. Each Ardupilot instance uses a different TCP port. Therefore, one DUNE module must be started for each UAV, configured with the correspondent TCP port. Finally, Neptus [21], a command and control software, is used to visualize the UAVs telemetry and location and to give commands to the UAVs, such as take off, loiter and to start/stop the Search and Rescue mission.

### 3.2. Mission Simulation Scenario and Parameters

In this section, the parameters that define the mission scenario are described.

#### 3.2.1. Aircraft and Remote Sensing Platform

The X8 is a battery powered small UAV which can fly for around 80 min with the automated remote sensing payload and proper battery. The radius of the remote sensor’s range is equal to 200 m, which is half of the width of the sensor’s footprint. This footprint was chosen assuming that a computer vision algorithm, such as the one described by [31], can detect the target in images captured at 400 m of altitude by an infrared camera with 7.5 mm of lens focal length, 640×480 pixels of resolution and 17 μm of pixel size.

#### 3.2.2. Search Domain

The reference search area used in this work is equivalent to the Search Effort ($Z_{a}$) of one X8 UAV, calculated by Equation (Equation 16). Considering the total endurance of 80 min, the On-Scene Endurance (*T*) is equal to 60 min (85% of the total endurance). The Search Facility Speed (*V*) is equal to the average airspeed of the aircraft, in this case 16 m/s. The Sweep Width (*W*) is equal to 400 m, which is the lateral length of the required sensor footprint. Therefore, the search area is equal to 23.04 km^2^, which gives a length and width equal to 4.8 km as the area has a squared shape because the datum is a single point.

#### 3.2.3. Cells Grid

The grid was built with a cell width of 100 m. Therefore, the 23.04 km^2^ of search area were divided into 48×48 cells. A two-dimensional normal distribution curve was fitted to the 12×12 single point datum reference table provided by the IAMSAR manual [27]. The fitted curve of the probability (Figure 8) that gives the reward of each cell is given by:(22)$$\phi_{mn}=0.002946\exp\left(-\left(\frac{{\left(m-23.5\right)}^{2}}{108.28}+\frac{{\left(n-23.5\right)}^{2}}{108.28}\right)\right),$$
where *m* and *n* are the horizontal and vertical indexes of the cell, respectively.

#### 3.2.4. MPC Parameters

The airspeed range was chosen to be between 12 to 22 m/s. The reason for this choice was to keep the airspeed around the cruise speed, so that the battery consumption did not reach values that were too high. The roll angle range was chosen to be between –45 and 45 degrees so that the aircraft performed smoother maneuvers but still with freedom. The safe distance between UAVs was chosen to be 100 m and a wind of 9.9 m/s pointing to 45 deg was considered in the flight dynamics simulation.

A time horizon of 20 s and 20 horizon steps were the parameters chosen for the MPC problem. With, for example, a ground speed of 17 m/s, this means 340 m of straight distance, or a 180 deg turn. The weighting factor *a* was chosen to be 10,000 because the rewards are of a very low value (the sum of all cells rewards is equal to one). The weighting factors *b* and *c* were chosen to be 1, in order to avoid unnecessary aggressive maneuvers.

Regarding the PSO parameters for the optimization, a total of 384 particles was used and the algorithm runs 35 iterations with local and global coefficients of 1.

#### 3.2.5. Simulation Platform

The optimization algorithm was written in CUDA C programming language in order to benefit from the parallelism, with the goal to embedded it on a NVIDIA^®^ Jetson^TM^ board in the future for Harware-In-The-Loop simulations and field tests.

The simulations were run in a laptop with the NVIDIA 940MX graphics card, which has 384 CUDA cores and 8 GB 128-bit memory. With the parameters presented in this study, each UAV is able to run one optimization in around 400 ms when three UAVs perform the optimization at the same time. When only one UAV is used, each optimization loop takes around 250 ms. Therefore, it is expected that the NVIDIA^®^ Jetson^TM^ board, which has 256 CUDA cores and 4 GB 64-bit memory will be able to run one optimization in less than 400 ms. If this performance is not achieved, the parameters can be fine-tuned to achieve shorter processing time. Another possibility is to implement an optimization stopping feature that will run as many iterations as possible within a given time, instead of a fixed number of iterations.

It is also relevant to mention that the optimization time of each step was adjusted to 400 ms also when using only one or two UAVs. This was done by inserting a delay, so each UAV’s optimization time is the same for simulations when one, two or three UAVs are used. Therefore, this gives a fair comparison of the results.

### 3.3. Simulations

Three operational profiles were evaluated for the mission scenario: employing only one UAV; employing two UAVs; or employing three UAVs. Five missions were executed for each one of the profiles in order to obtain the average performance.

The reference search area was the Total Adjusted Search Area ($A_{t}$) for one UAV facility and Probability of Detection (POD) equal to 1, as described in the previous section.

The area was kept the same when employing two or three UAVs in order to allow a fair performance comparison between the profiles. Figure 9 illustrates one mission with three UAVs being monitored by the Command and Control software Neptus. The light red area is the search area and the dark red cross in the middle is the single point datum.

In all missions, the UAVs departed from the same region (southeast of the search area as shown in Figure 10) where they were loitering and waiting for the mission start command. After receiving the command, the UAVs departed to the search area and the mission time started to count from when the first UAV collected the first reward.

The IAMSAR manual describes the Probability of success (POS) as the probability of finding the search object with a particular search. For each sub-area searched, POS=POD×POC. It is therefore the way to measure search effectiveness. As the Probability of detection (POD) is kept at 1, the POS is equal to the POC of the area, which in this work is the sum of all rewards collected by the group of UAVs in the area.

### 3.4. Results

The boxplot of the time to reach 50% of POS is shown in Figure 11 for the three operational profiles: employing one, two, or three UAVs. It is possible to notice that the gain when a pair of UAVs is used is very significant when compared to the single UAV profile, reaching 50% of POS 84% faster. When adding a third UAV, the gain was less significant: on average, the group reached 50% of POS 28% faster than when employing a pair of UAVs. The decrease on the gain is probably due to the fact that the UAVs are often flying over areas that have already been flown. A possible solution to avoid this situation is to reduce the width of the cells, increasing the resolution of the grid. Therefore, the UAVs would better tune their maneuvers and still have the cells inside the UAVs’ sensor radius. However, this will increase the required computational power. This issue could be mitigated by optimizing the algorithm, for example.

Figure 12 shows the average POS during 20 min of mission. It is possible to notice that the results match the observed behavior when the missions were monitored. From the mission start to around 4 min, the UAVs fly to the area close to the datum, where the reward (Probability of containment) is higher. When two UAVs are employed, they fly parallel so that they cover more cells than when employing a single UAV. However, when three UAVs are employed, even if they form a parallel path, they fly close to each other and, therefore, do not visit many more cells than the pair of UAVs. This happens because, in case the three UAVs got far enough from each other to not visit the same cells, they would take a longer path to arrive at the central area (highest rewards), not being a cost beneficial solution.

After reaching the area close to the datum at around 4 min, the curve of rewards collection grows steadily and the difference between the three operational profiles is clear. When three UAVs were employed, the group reached 50% of POS 2.35 times faster than the single UAV. The gain, however, reduces over time. For example, to reach 65% of POS, the group of three UAVs did it 2.07 times faster than the single UAV. The reason for the decrease on the gain is that the more cells are already visited and rewards collected, the further the UAVs have to fly to visit new cells and collect new rewards (that are also lower in value). Therefore, the closer it is to the end of the missions, the smaller is the difference between the performance of the different operational profiles.

Figure 13 shows the boxplot of the POS after 20 min of mission for the three operational profiles. According to the IAMSAR Manual calculations, the single UAV is expected to reach 100% of POS in 60 min. It is possible to notice that the group of three UAVs is able to reach close to 90% of POS in 20 min, showing that the improvement of adding extra UAVs is approximately linear.

In order to evaluate the benefits of using the wind information in the kinematic model and test the system’s robustness, extra simulations were performed for: a scenario where the wind was not considered in the MPC; and a scenario where the wind considered in the MPC was underestimated by 20%. Results are shown in Figure 14. It is possible to notice that the performance is superior when the wind information is used, collecting on average 4.4% more rewards in 20 min of mission time than when the wind is not considered in the model. The performance of the underestimated wind case was close to the ideal case, proving the system’s robustness. It is important to notice that, despite the higher median of the underestimated wind case, the ideal case was able to reach higher values of rewards collected having both the upper quartile and maximum value higher than the underestimated wind case.

Finally, a pre-made path was created (Figure 15) in order to compare to the performance of the single UAV with the real-time MPC optimization. In this path, the UAV flies from the mission origin to the grid’s midpoint and then flies a spiral path. This spiral path is close to the standard path suggested by the IAMSAR Manual.

In the spiral path, the lanes are equally spaced allowing the best coverage by the sensor’s footprint. This would be the best possible simple standard path for the mission scenario being investigated. In addition, before starting the spiral path, the UAV is assumed to first fly to the center of the area.

Figure 16 shows that, for the same average airspeed of 15.5 m/s, the performance of the MPC path-planning was superior in the first 20 min of mission. In addition, in the spiral path, 50% of POS was reached in around 13 min, while it took less than 11 min when the MPC optimization was used.

In the spiral path, wind was not considered and the UAV keeps the ground speed constant, while, in the MPC path-planning, the UAV optimizes its speed to reach higher coverage, for example reducing the airspeed to achieve a steeper turn when needed. Another advantage, which is perhaps the most important, is that the MPC solution has the capability to deal with dynamic changes on the environment and mission parameters during the mission, as it is a real-time optimization. These changes can be wind variations, updated search and rescue reports or even the lost of one UAV in the middle of the mission due to technical problems or the addition of extra UAVs that arrived later when the mission had already started.

## 4. Conclusions

In this article, a real-time path-planning for search and rescue with Model Predictive Control solved by Particle Swarm Optimization was proposed. The solution was implemented to be embedded in the UAVs’ on-board computer and tested in a Software-In-The-Loop environment with flight dynamics simulations. In future work, Hardware-In-The-Loop simulations will be conducted in order to prepare the system for flight tests. The search area was defined using the International Aeronautical and Maritime Search and Rescue (IAMSAR) directives. In addition, the area was divided into a grid of cells, where each cell had a correspondent reward, referred to the IAMSAR’s Probability of containment. Results were analyzed for missions where one, two or three Unmanned Aerial Vehicles (UAVs) were employed. To reach 50% of Probability of success, the performance of the group of three UAVs was on average 2.35 times faster than the single UAV search. The effects of the inclusion of wind information were also evaluated. When wind information was used, even if the wind speed was underestimated by 20%, the team of three UAVs achieved on average 4.4% higher Probability of success in 20 min of mission, when compared to the scenario where no wind information was used. The performance of the single UAV was also compared to a standard search pattern based on the IAMSAR’s suggested pattern. The search using the proposed solution outperformed the standard search pattern in the first 20 min, with the additional advantage of being a real-time method that can deal with environmental dynamic changes and new mission directives.

## Figures and Tables

**Figure 1 sensors-19-04067-f001:**
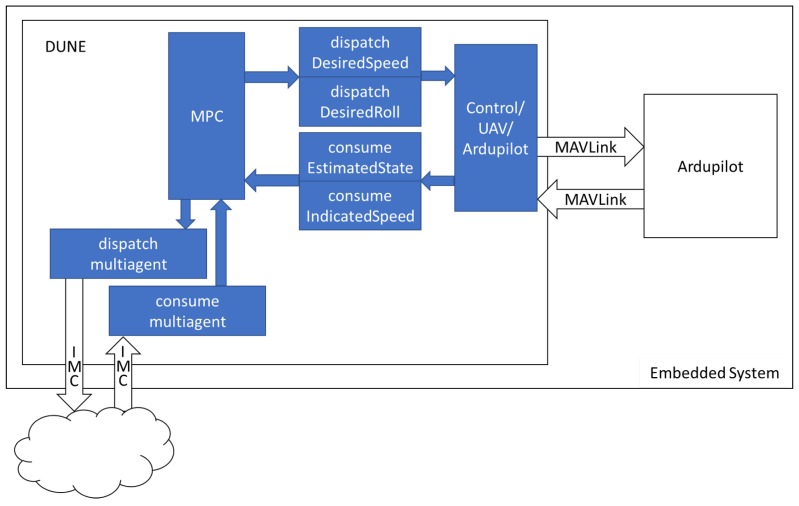
Simplified DUNE block diagram.

**Figure 2 sensors-19-04067-f002:**
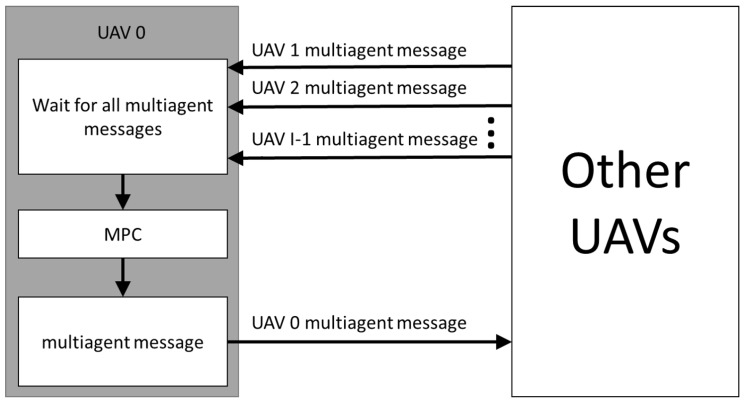
Unmanned Aerial Vehicle (UAV) agents flowchart.

**Figure 3 sensors-19-04067-f003:**
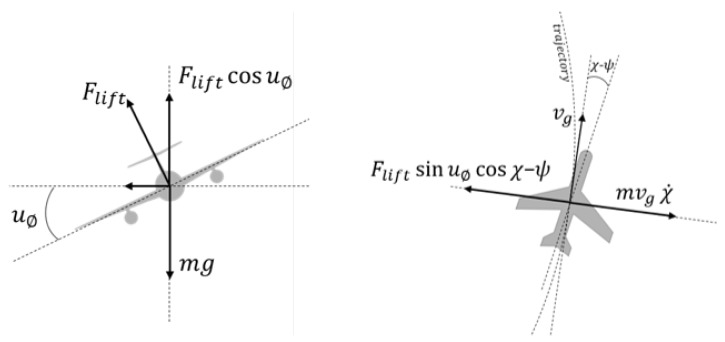
Vertical (**left**) and horizontal (**right**) forces in the coordinated turn maneuver.

**Figure 4 sensors-19-04067-f004:**
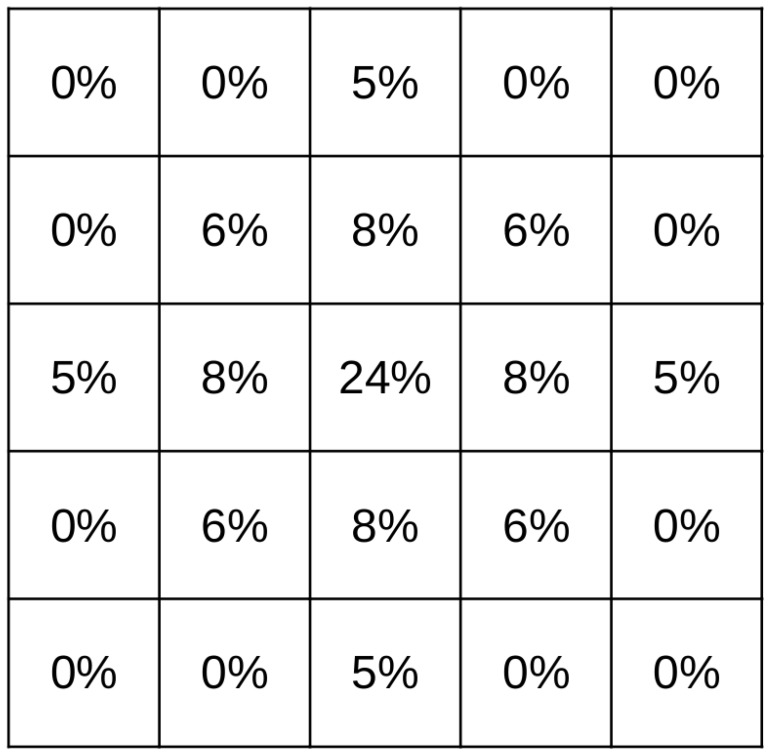
Initial probability table example.

**Figure 5 sensors-19-04067-f005:**
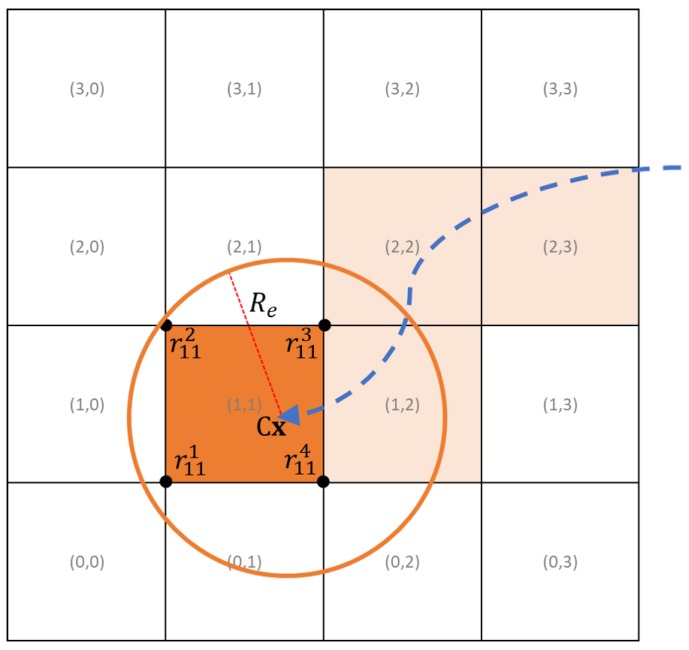
Cells grid example.

**Figure 6 sensors-19-04067-f006:**
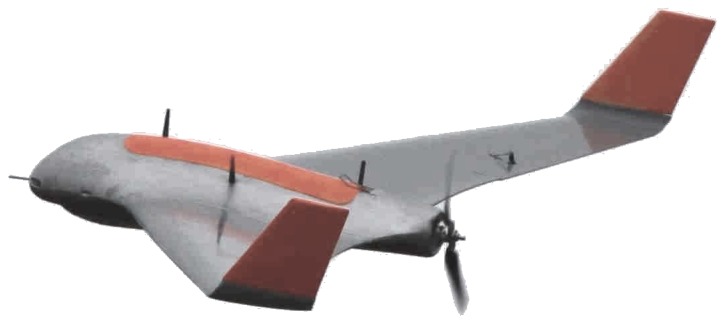
X8 UAV. Source: NTNU.

**Figure 7 sensors-19-04067-f007:**
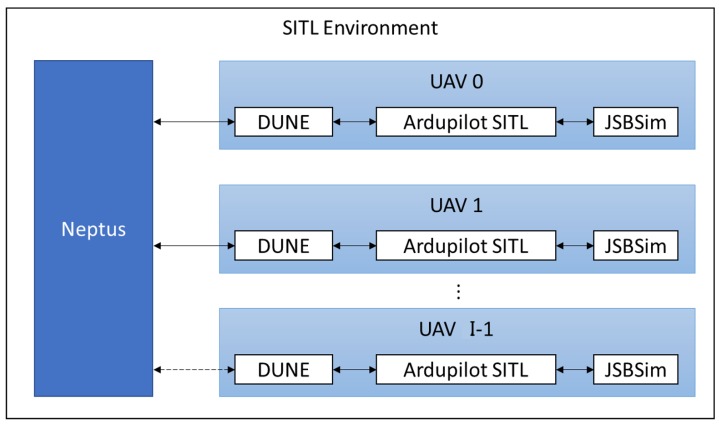
Software-In-The-Loop setup.

**Figure 8 sensors-19-04067-f008:**
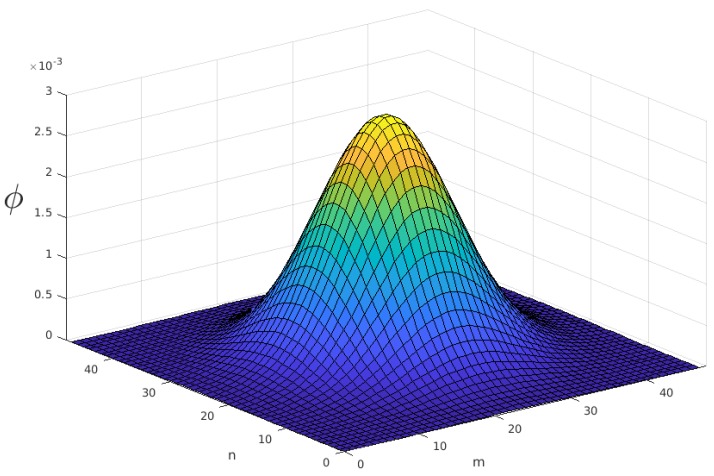
Reward of cells.

**Figure 9 sensors-19-04067-f009:**
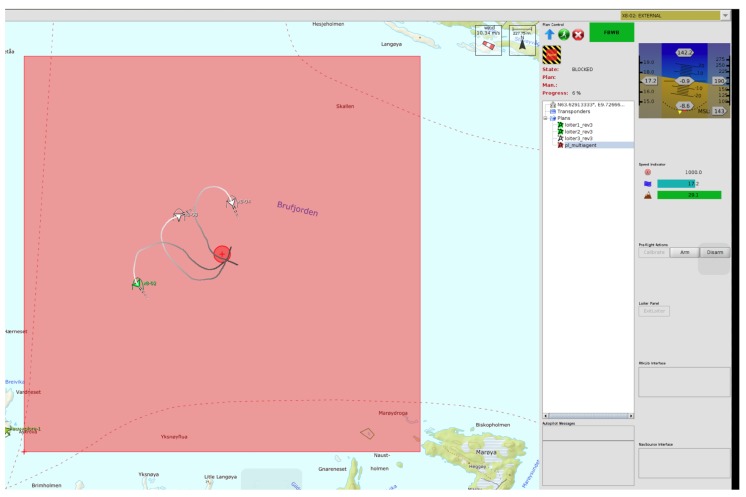
Snapshot of a mission with three UAVs being monitored with Neptus.

**Figure 10 sensors-19-04067-f010:**
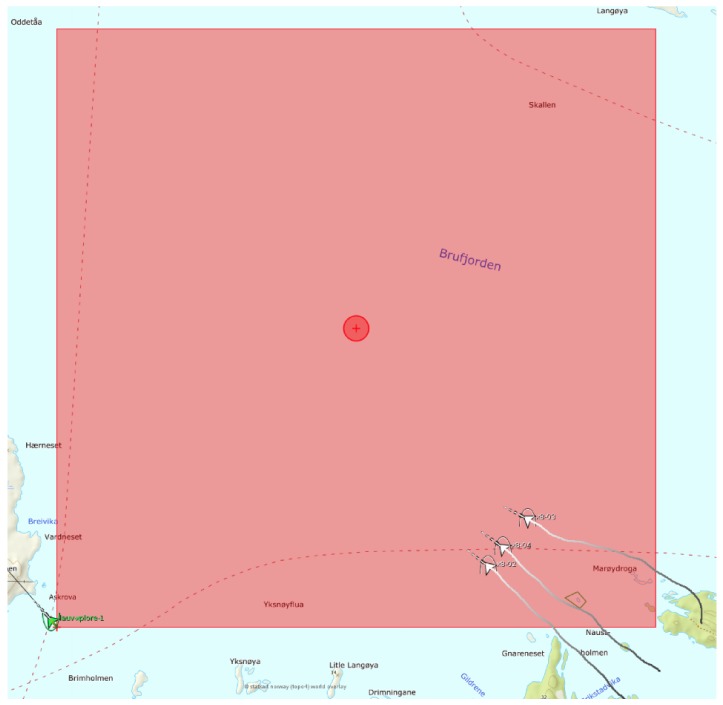
Snapshot of the beginning of a mission.

**Figure 11 sensors-19-04067-f011:**
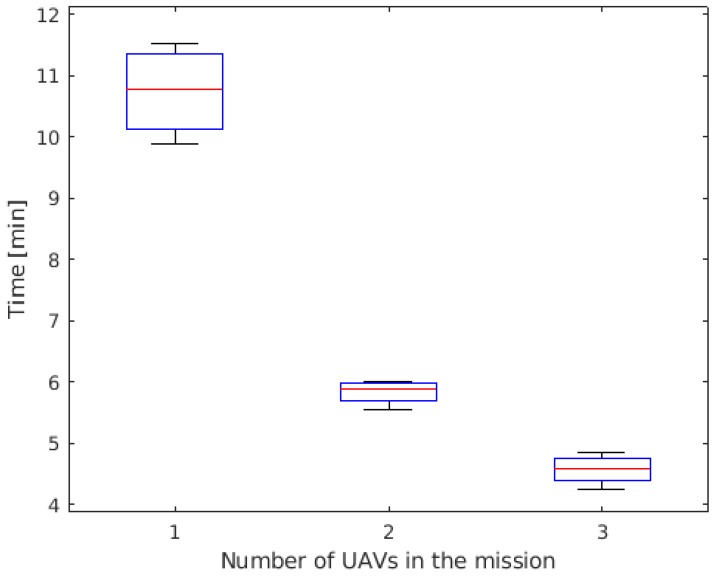
Time to reach 50% of Probability of success (POS).

**Figure 12 sensors-19-04067-f012:**
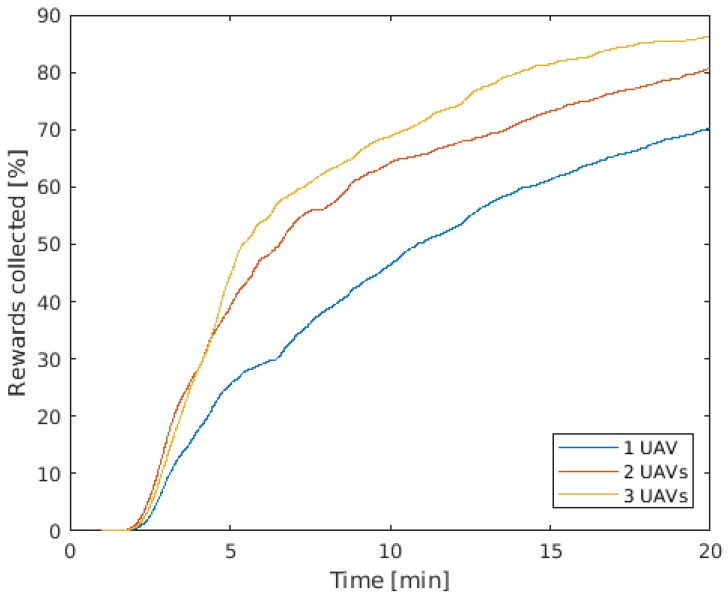
Average Probability of Success in time.

**Figure 13 sensors-19-04067-f013:**
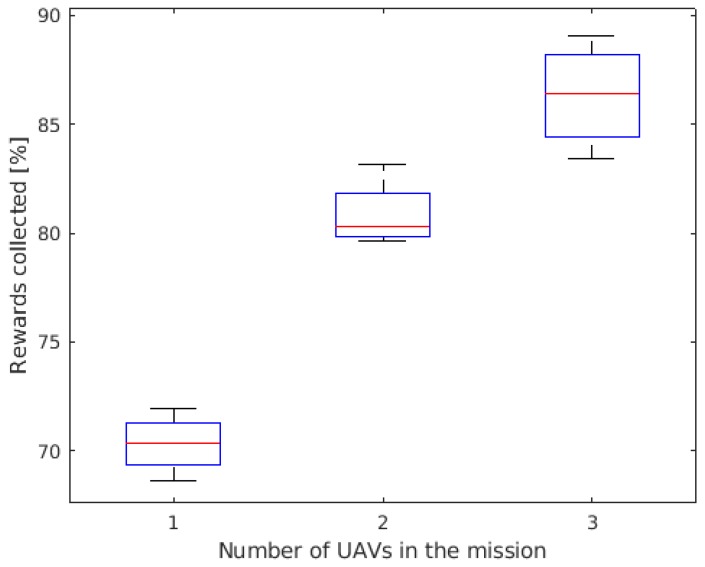
Probability of success (POS) in 20 min of mission.

**Figure 14 sensors-19-04067-f014:**
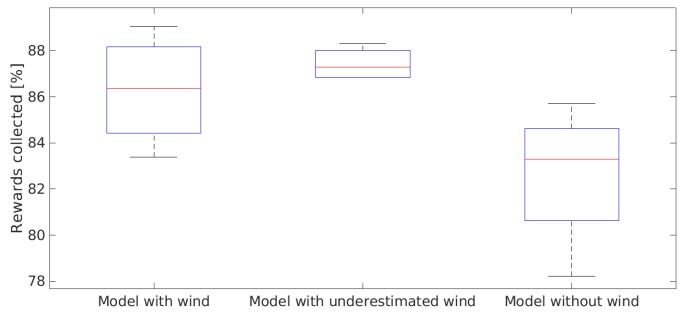
Probability of success (POS) in 20 min of mission.

**Figure 15 sensors-19-04067-f015:**
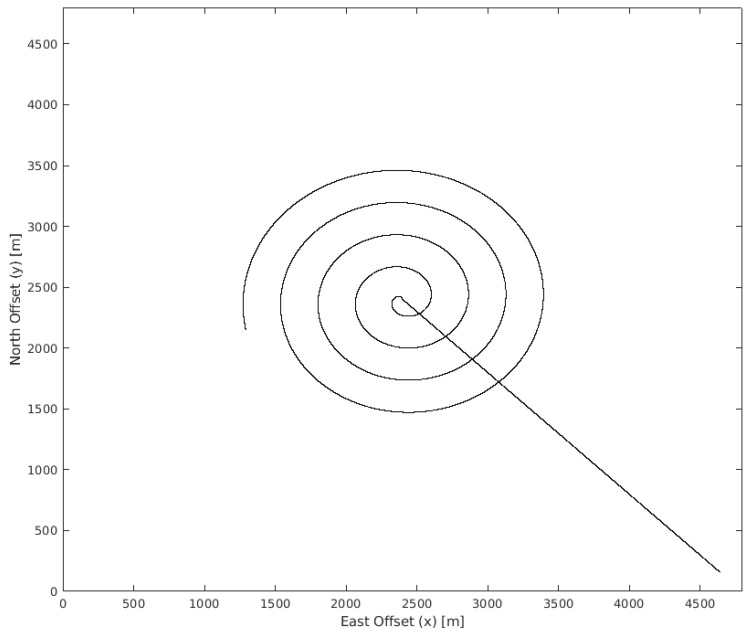
Spiral path.

**Figure 16 sensors-19-04067-f016:**
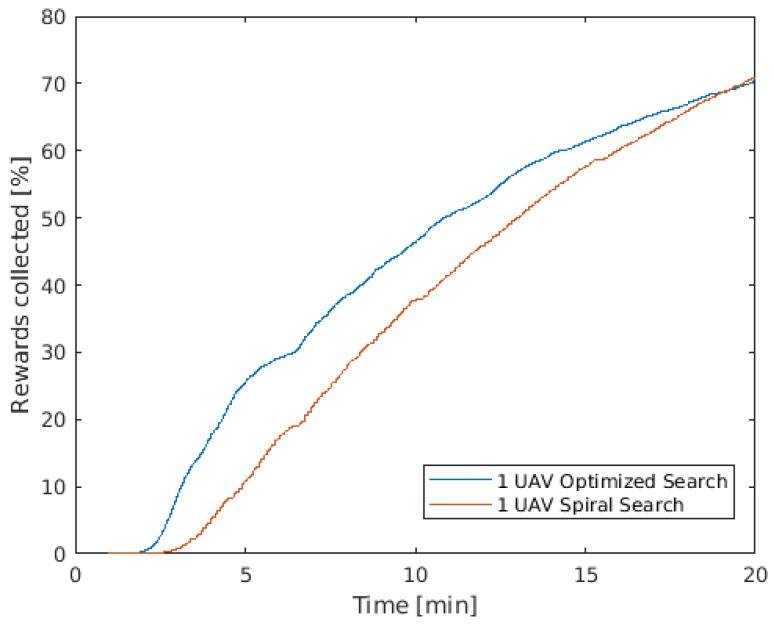
Probability of Success.

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
