# Peer review of "Autonomous Unmanned Aerial Vehicles in Search and Rescue Missions Using Real-Time Cooperative Model Predictive Control"

_sensors, 2019, doi:10.3390/s19194067_

Round 1
Reviewer 1 Report
A multi-UAV SAR system based on MPC is developed in this paper. I think the topic is interesting and the designed system has a good application prospect. But there are some problems that the authors should concern and revise them carefully as follows:
The authors should illustrate their innovations item by item in Introduction. The wind is introduced in the designed system. What is the detailed influence of the wind on the path planning process? How can the proposed algorithm address it? What are the differences between the numerical simulation and the SITL simulation in SAR missions? What is the necessity of the introduction of the SITL simulation? In Lines. 78-80, the authors claim that communication delays, processing time, actuator limitations are challenges for the SAR path planning. How do the authors address these challenges in their developed system? What are the adopted guidance law and flight control law in the developed system? The authors can briefly introduce them. The future work can be supplemented in detailed. The references are not sufficient enough in this paper. I suggest the authors can supplement some recent literatures about MPC/RHC applications in path planning and multi-UAV SAR missions. The following literatures can be referred:MPC/RHC applications in path planning:
(1) Energy management strategy for solar-powered UAV long-endurance target tracking. DOI: 10.1109/TAES.2018.2876738.
(2) Formation Obstacle Avoidance: A Fluid-Based Solution. DOI: 10.1109/JSYST.2019.2917786.
(3) Model predictive cooperative localization control of multiple UAVs using potential function sensor constraints. DOI: 10.1007/s1051.
Multi-UAV SAR missions:
(1) Distributed trajectory optimization for multiple solar-powered UAVs target tracking in urban environment by Adaptive Grasshopper Optimization Algorithm. DOI: 10.1016/j.ast.2017.08.037.
(2) LSAR: Multi-UAV Collaboration for Search and Rescue Missions. DOI: 10.1109/ACCESS.2019.2912306.
(3) Intelligent UAV map generation and discrete path planning for search and rescue operations. DOI: 10.1155/2018/6879419.
Author Response
Reviewer comment: A multi-UAV SAR system based on MPC is developed in this paper. I think the topic is interesting and the designed system has a good application prospect. But there are some problems that the authors should concern and revise them carefully as follows: Authors comment: Thank you very much for the comments. See below our comments regarding the improvements that we introduced in the new draft. Reviewer comment: The authors should illustrate their innovations item by item in Introduction. Authors comment: We used the text of the two last paragraphs of the introduction to include an item by item description of the innovations, as suggested. Reviewer comment: The wind is introduced in the designed system. What is the detailed influence of the wind on the path planning process? How can the proposed algorithm address it? Authors comment: We included extra results to show the influence of the wind in the path planning process. In the new results, the performance of the system is compared for a scenario where the wind is not considered in the MPC and for a scenario where the wind was underestimated by 20%. The wind information is assumed to be known. It can be estimated with on-board sensors or gotten from forecasts. The wind effects are included in the kinematic model, which is used by the MPC algorithm. Reviewer comment: What are the differences between the numerical simulation and the SITL simulation in SAR missions? What is the necessity of the introduction of the SITL simulation? Authors comment: SITL simulations are numerical simulation that runs in real time and closed loop with the control system software (in our case the Ardupilot). We included in the text a more detailed explanation of how the SITL environment works and the reasons for using it. The necessity of the introduction of SITL simulations is to test the integration of the proposed algorithm to the UAV embedded software (such as DUNE and Ardupilot) and also to analyse the behavior of the UAV using flight dynamics simulations. We tried to make this more clear in the text also. Reviewer comment: In Lines. 78-80, the authors claim that communication delays, processing time, actuator limitations are challenges for the SAR path planning. How do the authors address these challenges in their developed system? Authors comment: We included a paragraph explaining how the system addresses communication and processing delays. Regarding the effects of the processing/actuator/communication issues, they are analysed through the SITL simulations. Reviewer comment: What are the adopted guidance law and flight control law in the developed system? The authors can briefly introduce them. Authors comment: We included a more detailed description of the guidance law used to develop the coordinated turn model. Regarding the flight control law, we included an explanation about the flight mode being used by the autopilot, which is responsible for the low level control inputs. Reviewer comment: The future work can be supplemented in detailed. Authors comment: We included in the conclusions a text about future work mentioning Hardware-In-The-Loop simulations and field tests. Reviewer comment: The references are not sufficient enough in this paper. I suggest the authors can supplement some recent literatures about MPC/RHC applications in path planning and multi-UAV SAR missions. The following literatures can be referred: MPC/RHC applications in path planning: (1) Energy management strategy for solar-powered UAV long-endurance target tracking. DOI: 10.1109/TAES.2018.2876738. (2) Formation Obstacle Avoidance: A Fluid-Based Solution. DOI: 10.1109/JSYST.2019.2917786. (3) Model predictive cooperative localization control of multiple UAVs using potential function sensor constraints. DOI: 10.1007/s1051. Multi-UAV SAR missions: (1) Distributed trajectory optimization for multiple solar-powered UAVs target tracking in urban environment by Adaptive Grasshopper Optimization Algorithm. DOI: 10.1016/j.ast.2017.08.037. (2) LSAR: Multi-UAV Collaboration for Search and Rescue Missions. DOI: 10.1109/ACCESS.2019.2912306. (3) Intelligent UAV map generation and discrete path planning for search and rescue operations. DOI: 10.1155/2018/6879419. Authors comment: We included the three SAR references and the first MPC reference, which were very related to the solution proposed in our paper. The first SAR reference also uses cooperative MPC and was particularly interesting.Reviewer 2 Report
As attached.

Author Response
Authors comment: Thank you very much for the review. Please see below how we addressed each comment.
Major Comments:
Reviewer comment: 1. Lack of contribution and significance
In the introduction part, the author applied MPC to SAR mission since MPC introduces the dynamic constrains ( Line 45 to 46). However, in control system design, the weight factor b and c in Eq.(11) were set to zero. The author explained that and (Line 321 to 323). This is hard to understand why MPC was chosen in this study instead of other optimal control methods. Also, the Eq.(11) can be written with only the first term and Eq.(7) and (8) are meaningless since u_k actually is not introduced in the cost function. As mentioned by the authors, the communication delays, processing time, actuator limitations are the challenges for real-life applications. And . I was expected to see some analysis of those problems. However, the actuator limitation was not discussed in this paper. Is there analysis on communication delay or processing time?
Authors comment: We redid the simulations including non-zero values for the weight factors b and c. Therefore, the aggressive maneuvers constraints are now correctly being addressed by the MPC in the cost function, avoiding unnecessary aggressive maneuvers. Regarding u_k, it was already introduced in the cost function in the form of the kinematic constraint (Eq. 9 in the new draft), so it was already being correctly considered in the optimal control problem. Communication, actuator and processing time issues are included by the SITL environment. Therefore, all these issues are present in the simulations and an evaluation of how the proposed solution addresses them is done. We rewrote the parts of the paper that mentioned actuator, processing time and communication challenges to make this more clear.
Reviewer comment: 2. The technical approach described in the paper needs further clarification. The authors used 2D kinematics model and control the vehicle with the desired airspeed and desired roll. What about the desired pitch angle in the simulation? Is there a fixed-pitch setpoint or the authors have revised the ardupilot software to simulate 2D case? What is the weight of the aircraft, wind area? etc.. These parameters are fatal to simulate the flight dynamics.
Authors comment: We included a description of the Ardupilot Fly-By-Wire-B flight mode used. In this flight mode, the Ardupilot is responsible for holding the UAV altitude. The altitude setpoint is fixed during the whole mission and only the commands of desired airspeed and roll angle are sent. The Ardupilot SITL simulations are 3D simulations with 3D flight dynamics simulated by JSBSim. We also included a more detailed description about the UAV platform and how the flight dynamics model was developed.
Reviewer comment: 3. The structure of the paper is hard for readers to follow: the whole control system structure was presented in section 2.4, while MPC controller was presented in 2.1.2, its cost function in 2.3 and the parameters are presented in 2.6.4. The system and structure can be presented first to give readers an overview and then explain the subsystem/functions of the system. The simulation platform and its setup details can be presented in Section 3, such as the wind setup, target etc.
Authors comment: We changed the structure to be more "top/down". First we show the whole system and then we explain the sub parts and functions. We also moved the explanation about the simulation to the results section.
Reviewer comment: 4. The abstract needs to be improved. The purpose and contribution are hard to figure out from the abstract.
Authors comment: We re-wrote parts of the abstract to emphasize the purpose and contributions.
Reviewer comment: 5. The authors showed the simulation results of their algorithm using a different amount of UAVs and one predesigned path. The algorithm was not compared with other optimal control methods, which made the conclusion weak. It will be much better to compare proposed method to at least one other optimal control method.
Authors comment: We believe that the purpose of the paper is to discuss a multiple UAVs system using MPC, which is considered state of the art and lacks of more studies discussing its implementation and results, especially in exploration problems. Also, the main advantage of a MPC solution is to deal with dynamic changes in the environment, what is not possible with off-line optimal control methods. The purpose of the paper is not to compare optimal control methods. Our focus is on the performance of the system for different numbers of UAVs in the team. As the international SAR manual suggests a spiral like path, we decided to include it to show a comparison with our system. In order to give to the readers more information about the performance of the proposed method, We now also include extra results for scenarios where the wind was not considered in the kinematic model or if the wind was underestimated.
Reviewer comment: 6. The authors emphasis their contribution to a real-time and embedded application. HITL simulation will be more convinced compared SITL simulation since the onboard component usually has limited computation power.
Authors comment: We agree. This is an interesting next step and we included a text about that as future work in the conclusions chapter. SITL simulation is an important phase of the development process, which makes us able to test the integration of the proposed algorithm to the UAV embedded software (such as DUNE and Ardupilot) and also to analyse the behavior of the UAV to the optimized controls using flight dynamics simulations. We tried to make this more clear in the text also. Regarding the computation power, we believe that our onboard component is able to run the optimization loop faster than the simulations presented in the paper. The computations are mainly done by the GPU. The simultaneous simulations for 3 UAVs done by the laptop were performed using 386 CUDA cores, so each UAV instance used 128 cores. A typical onboard component has 256 cores and is going to be used only for the calculations of one UAV instance.
Minor comments:
Reviewer comment: 1. Figure 3 can be removed since Figure 4 can explain the system structure as well.
Authors comment: We removed Figure 3.
Reviewer comment: 2. Figure 5 is hard to understand.
Authors comment: We redesigned the figure and improved the related text in order to make it easier to understand.
Reviewer comment: 3. The proposed method in this paper requires all the systems are in the same network. How to form the network if UAVs are in the air? The distance to ensure all vehicles are connected should also be one of the constrains.
Authors comment: We included a paragraph explaining that DUNE is responsible for the communication management and that we assume that within the area being searched the UAVs are always close enough for maintaining a communication link. Network issues are included in the future work for the Hardware-In-The-Loop simulations, which is the next step on the development of our solution. We believe that network issues are beyond the scope of the paper but are planned to be addressed in future work.
Reviewer comment: 4. The safety distance is determined based on the position in the NED frame. It will be more interesting to see safety distance in body frame since the fixed-wing aircraft cannot fly backwards and the velocity along body y-axis is much small than along body x-axis (suppose the x-axis extending toward the nose tip, the y-axis toward the rightwing, and the z-axis toward the back of the aircraft).
Authors comment: The distance between UAVs in the NED frame is calculated for each MPC horizon step. The states of the UAVs (position in the NED frame and course) are dependent on the control inputs and wind (which is assumed to be known). Therefore, the control inputs are adjusted so that the positions of the UAVs are far enough in order to keep the distance between them always higher than the chosen safety distance. Therefore, we believe that the safety distance constraint is well addressed.
Reviewer 3 Report
This paper proposes a real-time path-planning for search and rescue with Model Predictive Control solved by Particle Swarm Optimization. The authors should consider the following points to improve the paper in the final version:
Introduction Needs a definition of software in the loop and how it differs from hardware in the loop. It is confusing to the reader that is unfamiliar with the differences. Structure of the paper may imply (to some readers) that approach is implemented on an embedded system. A more explicit notice to readers may be beneficial. Model and Controller Formulation Some variables are missing subscripts. For example, after equation (11), it should be “ Consider u_v_k-1 and u_q_k-1” and not “u_v-1 and u_q-1”. The authors need to define x_k (state variables) and u_k (control inputs) after equation (4). The authors claim they considered the delay in the design. However, they didn’t mention how they consider the delay in the model or controller formulations. FiguresFigure 1 is too small. Please also provide the reference in the caption of figure 1. Figures 3 and 6 are not necessary. The arrows in Figure 5 are confusing. Recommend a redesign Experimental Setup In subsection 2.6.4., the authors mentioned they use the time horizon of 20 s and 20 horizon steps in the MPC problem. Is it 20 ms or 20 sec? (In the 2.6.5., you mentioned the optimization is run in around 400 ms) Please specify what are sampling time, horizon and control interval in the simulation. Need to include optimization run times for laptop. It is claimed that the NVIDIA Jetson TX2 can run the optimization in time, but there is no baseline time to compare against. Without it, there is no backing for this claim. Successful implementation on embedded hardware is one of the main purposes of this paper. English language and style small number of spelling and grammatical errors (e.g., after equation (17) “Is is” should be “It is”) A small number of sentences with awkward/unclear structure Improper structure of English idioms (e.g., "In the other hand" should be "On the other hand") Recommend a thorough read through to fix errors Please integrate the related paragraphs. For example, section 2.6.3, The first three sentences can be connected to be one paragraph.
Author Response
Reviewer comment: This paper proposes a real-time path-planning for search and rescue with Model Predictive Control solved by Particle Swarm Optimization. The authors should consider the following points to improve the paper in the final version: Authors comment: Thank you very much for the suggestions. See below how we addressed them. Reviewer comment: Introduction Needs a definition of software in the loop and how it differs from hardware in the loop. It is confusing to the reader that is unfamiliar with the differences. Authors comment: We included a definition of software in the loop and its advantages. We also included a text about future work that mentions hardware in the loop simulations. Reviewer comment: Structure of the paper may imply (to some readers) that approach is implemented on an embedded system. A more explicit notice to readers may be beneficial. Authors comment: We included more explicit notices about the fact that we developed the system with the goal to embed it on an embedded system and that the system is ready for that, however the simulations were done using a laptop. We also changed the structure of the paper to a more "top/down" structure. Reviewer comment: Model and Controller Formulation. Some variables are missing subscripts. For example, after equation (11), it should be “ Consider u_v_k-1 and u_q_k-1” and not “u_v-1 and u_q-1”. Authors comment: We reviewed the text and fixed missing super- and subscripts. Thanks. Reviewer comment: The authors need to define x_k (state variables) and u_k (control inputs) after equation (4). Authors comment: We defined them and also defined k that was missing. Reviewer comment: The authors claim they considered the delay in the design. However, they didn’t mention how they consider the delay in the model or controller formulations. Authors comment: There was a short piece about that. We expanded to a paragraph that explains how the delays are addressed. Reviewer comment: Figures. Figure 1 is too small. Authors comment: This figure is for illustrating the probability map example given by the IAMSAR manual. We made it a little bit bigger. Reviewer comment: Please also provide the reference in the caption of figure 1. Authors comment: Done. Thanks. Reviewer comment: Figures 3 and 6 are not necessary. Authors comment: We removed figure 3. We kept figure 6 since one of the reviewers asked us to write a little bit more about the UAV platform and its flight dynamics model configuration. Reviewer comment: The arrows in Figure 5 are confusing. Recommend a redesign Authors comment: We redesigned the arrows and tried to make the text related to the figure more clear. Reviewer comment: Experimental Setup In subsection 2.6.4., the authors mentioned they use the time horizon of 20 s and 20 horizon steps in the MPC problem. Is it 20 ms or 20 sec? Authors comment: The finite horizon is of 20 s, divided into 20 horizon steps. This gives a sampling period of 1 s. We rephrased this sentence in order to make it more clear. Reviewer comment: (In the 2.6.5., you mentioned the optimization is run in around 400 ms) Please specify what are sampling time, horizon and control interval in the simulation. Need to include optimization run times for laptop. Authors comment: The text was confusing and we fixed it by explaining that it is each optimization loop which takes around 400 ms. We also explained the optimization run times for the laptop. Reviewer comment: It is claimed that the NVIDIA Jetson TX2 can run the optimization in time, but there is no baseline time to compare against. Without it, there is no backing for this claim. Successful implementation on embedded hardware is one of the main purposes of this paper. Authors comment: We rewrote this subsection explaining that our onboard component is expected to run the optimization loop faster than the laptop because simultaneous simulations for 3 UAVs done by the laptop were performed using 386 CUDA cores, so each UAV instance used 128 cores. The onboard component has 256 cores and is going to be used for the calculations of only one UAV instance. Reviewer comment: English language and style small number of spelling and grammatical errors (e.g., after equation (17) “Is is” should be “It is”) A small number of sentences with awkward/unclear structure Improper structure of English idioms (e.g., "In the other hand" should be "On the other hand") Recommend a thorough read through to fix errors Please integrate the related paragraphs. For example, section 2.6.3, The first three sentences can be connected to be one paragraph. Authors comment: We reviewed spelling, grammar and paragraph integration. Thanks.Round 2
Reviewer 1 Report
I think this version can be accepted.
Author Response
Thank you!
Reviewer 2 Report
As attached.

Author Response
Thank you. See below our comments to the points.
Point 1 (lift compensation when rolling)
We know that there is increase of pitch or/and airspeed to gain lift when rolling. This is dealt by the low level controls of the Ardupilot. We included a sentence in the description of FBWB mode: "In this flight mode, the Ardupilot control unit is responsible for holding the aircraft's altitude and compensating the loss of lift caused by the rolling." In the FBWB mode, we choose the altitude using the elevator controller. We cannot control pitch. It would only be possible to include the pitch state in the model and calculate the pitch rate based on the lift loss in each time step. However, the model would not be a generic kinematic model but a dynamic model with the specific UAV platform parameters such as the lift coefficient. Also, we would need to know how the Ardupilot low level controls deal with the lift losses in order to correctly include it in the model equations. Beard's book "Small unmanned aircraft: Theory and practice" in the coordinated turn chapter says: "we can see that heading rate is related to the pitch rate, yaw rate, pitch, and roll states of the aircraft. Each of these states is governed by an ordinary differential equation. Physically, we know that heading rate is related to the roll or bank angle of the aircraft, and we seek a simplified relationship to help us develop linear transfer function relationships in coming sections of this chapter." Therefore, it is common for research to leave the increased pitch to compensate lift loss out of the coordinated turn kinematic model. Also, the effects are very small and including them in the model equations would bring computational complexity that is not worth the accuracy gain. For the UAV platform that we are using, wind tunnels (reference included in the paper) show that it is needed to increase only 2 deg of angle of attack when rolling 45 deg (max in the paper) in order to maintain the same lift of when flying leveled. This results in an insignificant effect on the position calculations as the heading rate is affected by a factor of sec(pitch), which would be 1.0006 for 2 deg of pitch. In order to avoid doubts about this, we included a sentence in the beginning of the coordinated turn section explaining that the effects of the increased angle of attack to compensate the lift losses when rolling are not considered in the kinematic model for simplicity and because the effects are not significant. Point 2 (values of b and c) The b and c weight factors are applied to the difference between the control inputs of consecutive time steps of 1 s. The time that the X8 UAV takes to switch from -45 to 45 degrees of roll is not very different from the time that it is able to change from 12 to 22 m/s of airspeed. Therefore, there is no problem that the weight factors have the same value. Also, the weight factors are used to avoid unnecessary aggressive maneuvers. I.e., if the same search performance is achievable without changing the control inputs, then they should not be changed. Therefore, the values of b and c don't matter much for the optimization convergence (as long as they are small and do not affect the search terms in the cost function). Point 3 (state errors in the cost function) The search problem is a high level control problem where the state errors are not explicitly formulated since a reference for the states is not explicitly formulated. The cost function is formulated in terms of search efficiency (similar to economic MPC) rather than some state error (as in classical MPC used for regulation or reference tracking). This is why state errors are not included in the cost function. Point 4 (Eq. 13) We fixed the equation. Thank you.